# Concentration Optimization of Localized Cu^0^ and Cu^+^ on Cu-Based Electrodes for Improving Electrochemical Generation of Ethanol from Carbon Dioxide

**DOI:** 10.3390/ijms23169373

**Published:** 2022-08-19

**Authors:** Hong Lu, Guan Wang, Yong Zhou, Aselefech Sorsa Wotango, Jiahao Wu, Qi Meng, Ping Li

**Affiliations:** 1School of Flexible Electronics (SoFE) & Institution of Advanced Materials (IAM), Nanjing Tech University, 30 South Puzhu Road, Nanjing 211816, China; 2Center of Excellence in Sustainable Energy, Department of Industrial Chemistry, Addis Ababa Science and Technology University, Amist Kilo, Addis Ababa 16417, Ethiopia; 3School of Physical and Mathematical Sciences, Nanjing Tech University, 30 South Puzhu Road, Nanjing 211816, China

**Keywords:** electrochemical CO_2_ reduction, Cu-based electrode, ethanol production, valence state regulation

## Abstract

Copper-based electrodes can catalyze electroreduction of CO_2_ to two-carbon products. However, obtaining a specific product with high efficiency depends on the oxidation state of Cu for the Cu-based materials. In this study, Cu-based electrodes were prepared on fluorinated tin oxide (FTO) using the one-step electrodeposition method. These electrodes were used as efficient electrocatalysts for CO_2_ reduction to ethanol. The concentration ratio of Cu^0^ and Cu^+^ on the electrodes was precisely modulated by adding monoethanolamine (MEA). The results of spectroscopic characterization showed that the concentration ratio of localized Cu^+^ and Cu^0^ (Cu^+^/Cu^0^) on the Cu-based electrodes was controlled from 1.24/1 to 1.54/1 by regulating the amount of MEA. It was found that the electrode exhibited the best electrochemical efficiency and ethanol production in the CO_2_ reduction reaction at the optimal concentration ratio Cu^+^/Cu^0^ of 1.42/1. The maximum faradaic efficiencies of ethanol and C_2_ were 48% and 77%, respectively, at the potential of −0.6 V vs. a reversible hydrogen electrode (RHE). Furthermore, the optimal concentration ratio of Cu^+^/Cu^0^ achieved the balance between Cu^+^ and Cu^0^ with the most favorable free energy for the formation of *CO intermediate. The stable existence of the *CO intermediate significantly contributed to the formation of the C–C bond for ethanol production.

## 1. Introduction

An electrochemical CO_2_ reduction reaction (CO_2_RR) uses electrical energy to convert CO_2_ molecules into hydrocarbon products, which can store the intermediate electricity as chemical energy in the form of fuels [1]. Various products, such as carbon monoxide [2,3,4], formic acid [5,6], methane [7], methanol [8], ethylene [9,10], acetic acid [11], ethanol [12,13], and minor C_3+_ products [14], have been acquired from direct and homogeneously catalyzed CO_2_ reduction. The selectivity of a specific product of the CO_2_RR is heavily dependent on the electrocatalysts. Generally, Zn [4], Ag [2], and Pd [3] metals are extensively explored to produce CO, while Sn [5], Bi [5], and Pb [6] metals have been studied to generate HCOOH from the CO_2_RR. Other high-value products containing alcohols and olefins are difficult to obtain from the CO_2_RR owing to their multi-electron requirement, which is provided by the surface atomic arrangement of electrocatalysts. A significant barrier to this conversion is the lack of efficient and robust catalysts for CO_2_ reduction, particularly the catalysts that can realize high-order products such as ethanol, methanol, or multi-carbon compounds.

Among conventional electrocatalysts, Cu-based electrodes can effectively enhance CO_2_ conversion into two-carbon products, such as ethanol and ethylene, owing to their unique electron energy density [15,16]. Simultaneously, the appropriate binding energy between the Cu electrode and intermediate *CO can stabilize protonated intermediates to inhibit the CO gas generation, which would poison the electrode [17]. For the CO_2_RR, Cu-based electrodes have higher selectivity to ethylene than ethanol owing to its lower energy barrier [18]. This implies that for Cu-based electrocatalysts with different reaction intermediates, ethanol production requires strict thermodynamic and dynamic conditions [19]. Correspondingly, the specific intermediate generated during the reaction determines the final products [20,21]. For example, *CO is generally considered to be the common initial intermediate for products such as alkanes and alcohols. An appropriate amount of adsorbed *CO species can be dimerized to CO–CO, which is the key intermediate for C–C coupling and C_2_ products formation, e.g., *C_2_H_x_O (x = 0–4) compounds [22]. Sufficient amounts of *CO on the surface of the electrocatalyst shortens the distance of C–C coupling, effectively improving the generation of two-carbon compounds. Moreover, the abundant *CO distribution is beneficial for protonation, helping to produce ethanol by lowering the energy barrier of ethanol formation [23]. For Cu-based electrodes, fine-tuning the electrode interface provides an opportunity to improve C–C coupling for the desired product. In particular, Cu-based materials with mixed valence states of Cu are advantageous for the C–C coupling in obtaining two-carbon products [24]. This is because mixed valence states form partial positively (*CO–Cu^+^) and partial negatively charged (*CO–Cu^0^) carbons, which result in the C–C coupling through electrostatic interaction. Cu^0^ undergoes chemical adsorption with the curved structure of CO_2_ owing to its full-shell electronic structure, and carbon atoms becomes negatively charged. At the same time, Cu^+^ combines H_2_O molecules with CO_2_ to form a hydrogen bond; the negative charge accumulates on the O atoms of CO_2_; the carbon atoms become positively charged [25]. The carbon atoms of two *CO on the Cu^+^ and Cu^0^ regions are positively and negatively charged, respectively, facilitating the C–C coupling. Shang et al. prepared a core–shell Cu catalyst (Cu@Cu_2_O), that is, a thin layer of Cu_2_O on the Cu surface, under environmental conditions. The synergistic effect between Cu^+^ and Cu^0^ on Cu@Cu_2_O increases ethanol yield and selectivity with faraday efficiency (FE) of 29% [26]. To produce ethanol efficiently, some researchers have also combined Cu compounds with other metal and semiconductor materials on the electrode interface. Cu@VO_2_ [27] and Au_x_/Cu_2_O [28] electrocatalysts were skillfully designed for ethanol generation, and their catalytic mechanisms were proposed. Molecular CO generated on Au or VO_2_ shifts to the adjacent Cu element, reducing the free energy of *COCO formation. Additionally, Cu plays an important role in the CO_2_RR. The correlation between ethanol production and the valence state of Cu should be urgently explored to enhance the CO_2_RR using a simple and effective approach.

In this study, Cu-based electrodes were prepared on fluorinated tin oxide (FTO) using the one-step electrodeposition method. Moreover, these electrodes were used as efficient electrocatalysts for CO_2_ reduction to ethanol. Monoethanolamine (MEA)—a complexing agent of the precursor Cu^2+^ ions—was used in the preparation process. The results showed that adding MEA precisely modulated the concentration ratio of Cu^0^ and Cu^+^ on the electrodes. This can probably be attributed to the complex formation between MEA and Cu^2+^ ions ([Cu(MEA)_2_]^2+^) in the precursor solution. On the one hand, the strong bonding effect between Cu^2+^ and MEA restrained the reduction of Cu^2+^ ions on the electrode. Likewise, the existence of [Cu(MEA)_2_]^2+^ inhibited the movement of Cu^2+^ ions to the electrode owing to its larger relative atomic mass than Cu^2+^ ions. These two factors simultaneously inhibited the reduction of Cu^2+^ ions and extended the existence time of Cu^+^ ions on the electrode. The spectroscopy characterization demonstrated that the concentration ratio of Cu^+^ and Cu^0^ on the Cu-based electrodes was controlled from 1.24/1 to 1.54/1 by regulating the amount of MEA. Moreover, the Cu-based electrodes prepared using MEA also significantly improved ethanol production, particularly at a lower voltage. For the Cu-based electrode containing the MEA volume ratio of 0.6% (Cu_0.6_), the maximum FE of ethanol was 48% and C_2_ was 77% at −0.6 V vs. RHE. The concentration ratio of Cu^+^/Cu^0^ was 1.42/1 for Cu_0.6_, and the electrode probably achieved the balance between Cu^+^ and Cu^0^ with a stabilized *CO intermediate. The abundant *CO intermediates contributed to the formation of the C–C bond for ethanol production.

## 2. Results and Discussion

### 2.1. Morphology of Cu-Based Electrodes

The Cu-based electrodes were prepared using copper nitrate and MEA as the reaction precursor, and the morphology and valence state agent, respectively, by constant potential electrodeposition at room temperature. In particular, Cu_0.6_ and Cu_1.2_ electrodes were obtained by adding MEA with a volume ratio of 0.6% and 1.2% to 0.1 mol/L copper nitrate, respectively. For comparison, a Cu-based electrode was fabricated without MEA using the same electrodeposition parameters (denoted as Cu_0_). Scanning electron microscopy (SEM) images showed that microparticles monodisperse on the electrode in isolation with a size of approximately 2 μm, and Cu_0_ consisted of several small nanoparticles with a diameter of 100 nm (Figure 1a and Figure 2c–f). These microparticles were grown on the Cu-based materials formed on the conductive layer of FTO (Figure 2a,b). In comparison, the Cu_0.6_ electrode possessed a flat surface densely arranged by smooth spherical particles of size approximately 300 nm, appearing as a uniform film (Figure 1b and Figure 3d–f). The size of the particles increased to approximately 500 nm for the Cu_1.2_ electrode, and these particles were decorated by smaller particles with approximately 50 nm diameter (Figure 1c and Figure 3j–l). Clear gaps between grainy particles with exposed substrates of bare FTO were observed (Figure 3a–c,g–i). The selective mapping in the 50 nm range showed that Cu and O elements were widely distributed on the plane for Cu_0.6_, and the distribution of O was mostly localized compared to that of Cu (Figure 1b,d–f). In the precursor solution, Cu^2+^ ions were evenly dispersed in the absence of MEA. Under the effect of the electric field, Cu^2+^ ions were aligned in the solution and then moved to the cathode. The chemical reduction of considerable Cu^2+^ ions improved the anisotropic growth of microparticles. Cu^2+^ ions, as a transition metal, are inclined to bond to MEA to form [Cu(MEA)_2_]^2+^ in the presence of MEA in the precursor solution [28]. The UV-vis absorption spectra of precursor solutions also demonstrated that the absorption edge of Cu^2+^ ions decreased and then the absorption peak of [Cu(MEA)_2_]^2+^ rose as the addition of MEA increased, which illustrated that the Cu^2+^ ions were made more complex by MEA with [Cu(MEA)_2_]^2+^ formation (Figure 4). Charged [Cu(MEA)_2_]^2+^ moved directionally to the cathode; however, the higher mass of [Cu(MEA)_2_]^2+^ slowed down its movement compared to Cu^2+^, resulting in smaller particles by suppressing grain growth for Cu_0.6_. However, the concentration of Cu^2+^ ions was reduced, and all of them were consumed after a large amount of MEA was added. Considerable amounts of [Cu(MEA)_2_]^2+^ delayed the crystallization of Cu-based materials, and insufficient Cu^2+^ ions led to cracking on Cu_1.2_. Atomic force microscopy (AFM) characterization further demonstrated that adding a suitable amount of MEA smoothed the surface of the Cu_0.6_ electrode so that it became flat, with the lowest surface roughness compared to those of the Cu_0_ and Cu_1.2_ electrodes (Figure 5).

### 2.2. The Phase Analysis of Cu-Based Electrodes

The X-ray diffraction (XRD) patterns of Cu-based electrodes clearly showed two peaks at 43.30° and 50.43°, corresponding to the (111) and (200) reflections of Cu (PDF#04-0836), respectively. The two peaks at 36.42° and 42.30° were attributed to the (111) and (200) reflections of Cu_2_O (PDF#05-0667) (Figure 6a), respectively. The Cu_0_ electrode exhibited stronger peaks of Cu, while the Cu_0.6_ and Cu_1.2_ electrodes exhibited the stronger peaks of Cu_2_O. To confirm the coexistence of Cu^+^ and Cu on a Cu-based electrode, an anodization experiment was conducted for the Cu_0.6_ electrode. Two typical oxidation peaks at 0.16 V and 0.07 V vs. RHE were detected (Figure 6b), which were attributed to the oxidation of Cu^0^ to Cu^2+^ and Cu^+^ to Cu^2+^, respectively [29].

### 2.3. Electrochemical CO_2_ Reduction for Cu-Based Electrodes

The performance of the CO_2_RR was evaluated in a 0.1 M KHCO_3_ saturated solution with potential ranging from −0.5 to −1 V (vs. RHE). Linear sweep voltammetry (LSV) was performed for the Cu_0.6_ electrode, and it was found that the Cu_0.6_ electrode exhibited a good activity for the electrochemical reduction of CO_2_ with a higher current in the CO_2_-saturated electrolyte than in N_2_-saturated electrolyte (Figure 7). Compared to Cu_0_, H_2_ generation was significantly inhibited on the Cu_0.6_ and Cu_1.2_ electrodes under all voltages. Moreover, ethanol generation was effectively improved at lower negative voltages, particularly for Cu_0.6_, which exhibited the most favorable selectivity of ethanol (Figure 8a–c). At the potential of −0.6 V vs. RHE, the FE of Cu_0.6_ for ethanol production was as high as 48%, which was optimal compared to those of most Cu-based electrodes (Table 1). For the Cu_0.6_ electrocatalyst, the yield of C_1_ products (CO and formic acid) under all voltages did not exceed 15%, and the FE of H_2_ was as low as 19%. Furthermore, the largest FE of C_2_ products was 77% for the Cu_0.6_ electrode at a potential of −0.6 V vs. RHE (Figure 8b and Figure 9a–c), which was attributed to its stronger C–C coupling capability consuming the intermediate *CO. However, the FE of the ethanol and C_2_ products of the Cu_1.2_ electrocatalyst was lower than those of Cu_0.6_ under all voltages, which might be due to the surface structure of the Cu_1.2_ electrocatalyst. In addition, the electrocatalytic performance of the Cu_0.3_ and Cu_0.9_ electrodes also verified that Cu_0.6_ is the best electrocatalyst for ethanol production, as shown in Figure 9d,e. The stability of Cu-based electrodes in CO_2_RR is of great significance for practical applications, considering the alkaline electrolyte and interface reliability. Continuous and stable operation of CO_2_RR electrolysis was implemented on the Cu_0.6_ electrode for 6 h under the voltage of −0.6 V vs. RHE. Moreover, no obvious changes in pH during the catalytic process (Figure 10a) was observed. The current remained above −0.5 mA cm^−2^ with a negligible decrease (Figure 10b–d), and the FE of ethanol was over 40% (Figure 8d) after 6h. The results of the SEM analysis of the Cu_0.6_ electrode after the CO_2_RR showed that the Cu-based material was well-maintained during the reaction (Figure 11a). Moreover, after the CO_2_RR, still two oxidation peaks existed for Cu_0.6_ as an anode, which suggested that the Cu_0.6_ electrode was stable during the electrocatalytic process (Figure 11b). Although the current density was slightly lower, it could be significantly improved by atomic rearranging and a composite strategy.

### 2.4. Electrochemical Characterization of Cu-Based Electrodes

The ultraviolet-visible-near-infrared (UV-vis) spectrometry showed strong absorption of MEA and [Cu(MEA)_2_]^2+^ at 200–320 nm, while no absorption for the Cu_0.6_ electrode (Figure 12a) was observed in the same wavelength range. This indicated the presence of a Cu-based material and the absence MEA or [Cu(MEA)_2_]^2+^ on the electrode. To study the dynamics of carriers, transient photoluminescence spectroscopy (TRPL) on Cu-based electrocatalysts was performed. Figure 12b shows that the fluorescence lifetime of Cu_0.6_ was 0.117 ns, which was lower than those of Cu_0_ (0.182 ns) and Cu_1.2_ (0.130 ns). The shorter lifetime of the carriers in the Cu_0.6_ electrode implied that the electrons quickly diffused to the interface, decreasing the non-radiative recombination of free electrons on the Cu-based electrodes. At the same time, the electrical impedance test also showed that the impedance of Cu_0.6_ was the smallest (Figure 12c), indicating that the charge transfer resistance was the lowest in the Cu_0.6_ electrode. The measurement of electrochemically active specific surface area (ECSA) indicated that the Cu_0.6_ electrocatalyst was capable of providing higher catalytic activity (Figure 13a–d). At the same time, the kinetic resistance of the catalyst for CO_2_RR was smaller for the Cu_0.6_ electrode with a smaller Tafel slope (Figure 12d). To explore the main cause of ethanol production for the Cu-based electrodes, we calcined the Cu_0.6_ electrode in Ar and air atmosphere (Figure 13e). For the Cu_0.6_ electrode post-treated in the Ar atmosphere, the obvious generation of ethanol (FE, 32%) and hydrogen (Figure 12d) was observed. However, no ethanol was detected for the Cu_0.6_ electrode temperature-treated in air, which was in an oxidized state without oxidation peaks (Figure 13f). The surface redox state of Cu was closely related to the CO_2_RR performance of the Cu-based electrodes.

### 2.5. X-ray Photoelectron Spectroscopy of Cu-Based Materials

Fine X-ray photoelectron spectroscopy (XPS) of Cu 2p in a Cu-based electrode was performed to analyze the valence state of Cu on the surface (Figure 14a,b). It is difficult to distinguish between Cu^+^ and Cu^0^ from the Cu 2p_1/2_ and Cu 2p_3/2_ peaks [30]. For Cu 2p_3/2_ and O 1s spectra, there was only one peak for each spectral line, which also demonstrated that the individual oxide of Cu_2_O exists in Cu-based material instead of CuO [31]. Further analysis with Cu LMM Auger energy spectrum (AES) demonstrated that Cu^+^ and Cu^0^ coexist on the electrodes (Figure 14c), and Cu^+^ ions are dominated at the peak of approximately 569.8 eV compared to 568.0 eV for Cu^0^ ions and 565.2 eV for the transition state of the Cu LMM [32]. These corresponding peaks were integrated to determine the ratio between Cu^+^ and Cu^0^ ions. For these as-prepared electrodes, the concentration ratio of Cu^+^ and Cu^0^ ions increased from 1.24/1 to 1.54/1 with increasing MEA content (Figure 14d–f). The strong interaction between Cu^2+^ and MEA affected the reduction of Cu^2+^ to Cu^+^ and further to Cu^0^ by Cu^+^ [33]. For the Cu_0.6_ electrode, the medium concentration ratio was 1.42/1, which showed the predominant CO_2_RR performance. The above AES characterizations were implemented on the flacking Cu-based powders. In addition, the AES of the electrodes was achieved, and it showed only the peak of Cu^+^ without Cu^0^ (Figure 15), which could be probably attributed to the localized distribution of Cu_2_O and Cu. Moreover, the electrode surface (0–5 nm depth) was covered by Cu_2_O. The product selectivity could probably be attributed to the synergy between Cu^+^ and Cu^0^ ions, which was demonstrated to accelerate CO_2_ activation and CO dimerization [26]

### 2.6. The Formation of *CO Intermadiate for Ethanol Production

The details of the valence state of Cu and its relationship with the *CO intermediate in CO_2_RR are worth exploring to obtain electrocatalysts with enhanced activity and high selectivity. The stable existence of the initial *CO intermediate played a significant role in C–C coupling, and it could be calculated using density functional theory simulations. The atomic structure models were implemented and *CO intermediates were bonded to the Cu atoms (Figure 16a–c). The degree of stability was represented by the free energy of the *CO intermediate. The free energy of *CO intermediate was −3.03 eV for Cu_0.6_, which was significantly lower than those for Cu_0_ (−1.85 eV) and Cu_1.2_ (−2.20 eV), indicating that the *CO intermediate was more stable on Cu_0.6_ than on Cu_0_ and Cu_1.2_. The stable existence of *CO significantly contributes to C–C coupling by the supply of the reaction precursor [34]. To date, there is no definite reactive process for CO_2_RR to ethanol. We propose one probable route considering the reported work (Figure 16d) [19,25,34,35]. The Cu_0.6_ electrode probably achieves the balance between *CO-Cu^+^ (containing positively charged C) and *CO-Cu^0^ (containing negatively charged C) [25]. The electrostatic attraction between these two C atoms contributes to the formation of the C–C bond. In addition to abundant *CO intermediates, adequately monoprotonated *CHO intermediates exist on the electrode, and they are prone to forming *CO-COH intermediates [34]. The CO–COH intermediates then accept multi-protons and electrons to obtain the precursor (*CH_2_CH_2_OH) of the final product ethanol [19,35]. The high specific surface area of Cu_0.6_ provides abundant reaction sites for the CO_2_RR and an appropriate Cu^+^/Cu^0^ ratio that accumulates reaction intermediates. The high concentration of the *CO intermediates on the Cu^0^-Cu^+^ interface further promotes the C–C dimerization reaction and improves the selectivity of ethanol.

## 3. Materials and Methods

*Synthesis of Cu-based electrodes on FTO:* The Cu-based electrodes were prepared using a one-step electrochemical deposition method on FTO. The three-electrode cell—Ag/AgCl as the reference electrode, FTO as the working electrode, and counter electrode—was used. Before the deposition, the bared FTO substrates were ultrasonically cleaned with isopropanol, ethanol, and water 6 times and quickly dried with nitrogen gas. Simultaneously, a definite volume of MEA (0, 60, 120, 180, and 240 μL) was dropped into a copper nitrate solution (0.1 mmol, 20 mL) as the deposited electrolyte. Then, the three-electrode cell was operated in the prepared electrolyte at a constant voltage of −0.4 V vs. RHE for 30 s, and five Cu-based electrodes were obtained in five precursor solutions with different volumes of MEA. The as-prepared electrodes were flushed with water three times and finally dried with nitrogen gas.

*Characterization:* The images were obtained using SEM (jsm-7800F, JEOL, Japan), and the SEM mapping was acquired using an energy-dispersive spectrometer in the same instrument. AFM analysis was conducted using an atomic force microscope (XE-70, Park systems, Korea). The XRD patterns of the samples were collected with a Smartlab (3 KW) X-ray powder diffractometer, and the Cu-K-α radiation wavelength was 0.154178 nm. Moreover, UV-vis spectra were collected using an ultraviolet spectrophotometer (PE Lambda 950, PerkinElmer, U.S.). TRPL analysis was conducted using a time-resolved fluorescence spectrometer (FLS 980, Edinburgh, UK). The XPS data were obtained using a K-alpha X-ray photoelectron spectrometer (PHI5000 Versaprobe, ULVAC-PHI, Japan). The pH of the solution was measured using Sartorius PB-10 (Sartorius, German).

*Electrochemical measurements:* An electrochemical workstation (CHI660E, Shanghai Chenhua, China) was used for electrochemical measurements. In the CO_2_RR characterization, the platinum electrode was used as the counter electrode, Ag/AgCl (saturated KCl) as the reference electrode, and the Cu-based electrode as the working electrode to form a three-electrode system. A proton exchange membrane (Nafion 117, Sigma-Aldrich, German) was inserted in the middle of electrolyte to ensure that only hydrogen ions could pass through the membrane (Figure 17). Further, 100 mL of 0.1 mmol KHCO_3_ solution was added to the reactor as the electrolyte with the remaining 150 mL of headspace volume. Before the reaction, high-purity carbon dioxide (99.99%) gas was vented into the electrolyte for 30 min to reach saturation.

All the potentials were converted to relative potentials according to the RHE reference value: $E\left(vs\;RHE\right)=E\left(vs\frac{Ag}{AgCl}\right)+0.197\;V+0.0592pH\;V.$

*Identification and quantification of gaseous products:* Gas chromatography (GC-9860-5C-NJ, Hao Erpu, China) was used to analyze the gas products, with argon (99.99%) as the carrier gas. A series of definite concentrations of gas were injected into the gas chromatograph (GC) to obtain the calibrated concentration of the gas products (H_2_, CO, CH_4_, C_2_H_4_, C_2_H_6_). Carbon-based gases and H_2_ were detected using a flame ionization detector and a thermal conductivity detector, respectively. Moreover, 1 mL of reactive gas was extracted each time and quickly injected into the GC for analysis.

*Identification and quantification of liquid products:* All the liquid products were quantified using a nuclear magnetic resonance spectrometer (JNM-ECZ400S/L1, JEOL, Japan). Different concentrations (0.5, 1, 1.5, 2, 2.5, 3, 3.5, and 4 mmol/L) of formic acid, methanol, acetic acid, and ethanol were prepared to obtain a correlation between the concentration and the peak intensity of the ^1^H spectra. Specifically, 400 µL of the above solution was mixed with 200 µL of deuterated dimethyl sulfoxide (d-DMSO, Adamas, German) nuclear magnetic sample, and presaturation was used for water suppression during the NMR spectrum test. The standard curve was obtained considering the product concentration and NMR spectrum. The product concentration was presented along the abscissa and the NMR peak intensity along the ordinate, with the deuterated peak as consultation (Figure 18).

The FE for the formation of all the products (both gas and liquid products) was calculated as follows: $FE=n\times e\times N_{A}\times \frac{q}{Q}=n\times e\times N_{A}\times q/\left(I\times t\right)$, where n is the total amount of product (in moles), e is the number of electrons transferred, N_A_ is the Avogadro constant, q is the elementary charge, Q is the charge, I is the current, and t is the running time.

*Computational method:* We have employed the Vienna ab initio simulation package (VASP) [36,37] to perform all density functional theory (DFT) calculations within the generalized gradient approximation (GGA) using the Perdew-Burke-Ernzerhof (PBE) [38] formulation. We have chosen the projected augmented wave (PAW) potentials [39,40] to describe the ionic cores and take valence electrons into account using a plane wave basis set with a kinetic energy cutoff of 450 eV. Partial occupancies of the Kohn−Sham orbitals were allowed using the Gaussian smearing method and a width of 0.05 eV. The electronic energy was considered self-consistent when the energy change was smaller than 10^−6^ eV. A geometry optimization was considered convergent when the force change was smaller than 0.03 eV/Å. Grimme’s DFT-D3 methodology [41] was used to describe the dispersion interactions. The Brillourin zone was sampled with a gamma-centered grid 2 × 2 × 1 through all the computational process. Periodic boundary conditions were used in all directions and a vacuum layer of 15 Å was used in the z-direction to separate the slabs.

The adsorption energy (Eads) of adsorbate molecule was defined as $Eads=E_{mol/surf}-E_{surf}-E_{mol}$, where *E_mol/surf_*, *E_surf_* and *E_mol_* (g) are the energy of adsorbate molecule adsorbed on the surface, the energy of clean surface, and the energy of isolated molecule in a cubic periodic box, respectively.

## 4. Conclusions

Cu-based electrodes were fabricated by a one-step electrodeposition method using MEA as a morphology and valence state agent. Moreover, the electrodes were used for electrochemical CO_2_RR. The addition of MEA smoothened the surface of the Cu-based electrodes and simultaneously modulated the valence state of the Cu element. The Cu_0.6_ electrode had a flat surface and a superior concentration ratio of Cu^+^ and Cu^0^ ions, which significantly improved the electrochemical efficiency and ethanol production in the CO_2_RR. For the Cu_0.6_ electrode, the maximum FE of ethanol was up to 48% at a potential of −0.6 V vs. RHE, which is also the largest value for single-material catalysts. At the same voltage, the overall C_2_ selectivity was above 77%. The flat surface significantly shortened the transfer time of electrons from the electrode to the reactive surface. At the same time, the Cu_0.6_ electrode exhibited the optimal Cu^+^/Cu^0^ concentration ratio of 1.42/1 compared to Cu_0_ and Cu_1.2_, which can stabilize the *CO intermediates with the lowest free energy. The synergistic effect of Cu^+^ and Cu^0^ is advantageous for the C–C coupling formed by the dimerization of carbon intermediates. The possible mechanism is that the C atoms bonding to Cu^+^ and Cu^0^ have opposite electrical charges, and the C–C bond is favorably formed under the effect of the electric field, that is, due to the electrostatic attraction. This study provides an efficacious guide to fabricating efficient electrocatalysts, which has potential significance for product selectivity in the CO_2_RR.

## Figures and Tables

**Figure 1 ijms-23-09373-f001:**
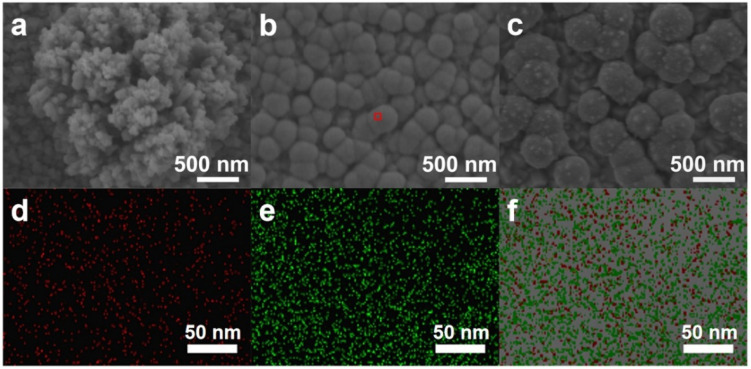
SEM images of the Cu-based electrodes: (**a**) Cu_0_, (**b**) Cu_0.6_, and (**c**) Cu_1.2_. Energy-dispersive spectrometry (EDS) mapping of Cu_0.6_: (**d**) O, (**e**) Cu, and (**f**) O and Cu. The selected range of EDS is from the area circled with red frame in (**b**).

**Figure 2 ijms-23-09373-f002:**
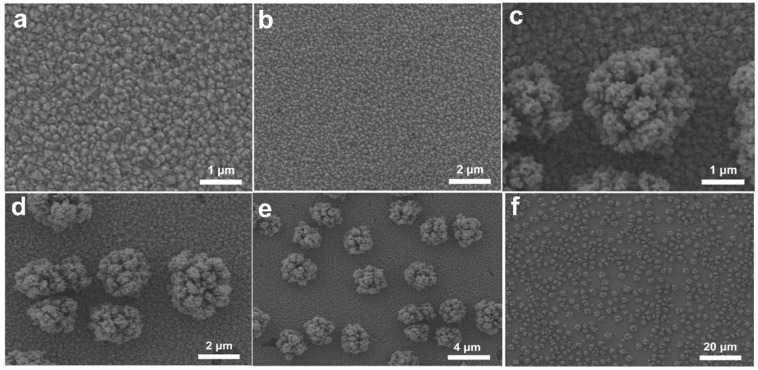
SEM images of the bared FTO with different scale bars: (**a**) 1 μm, (**b**) 2 μm. SEM images of Cu_0_ with different scale bars: (**c**) 1 μm, (**d**) 2 μm, (**e**) 4 μm, (**f**) 20 μm.

**Figure 3 ijms-23-09373-f003:**
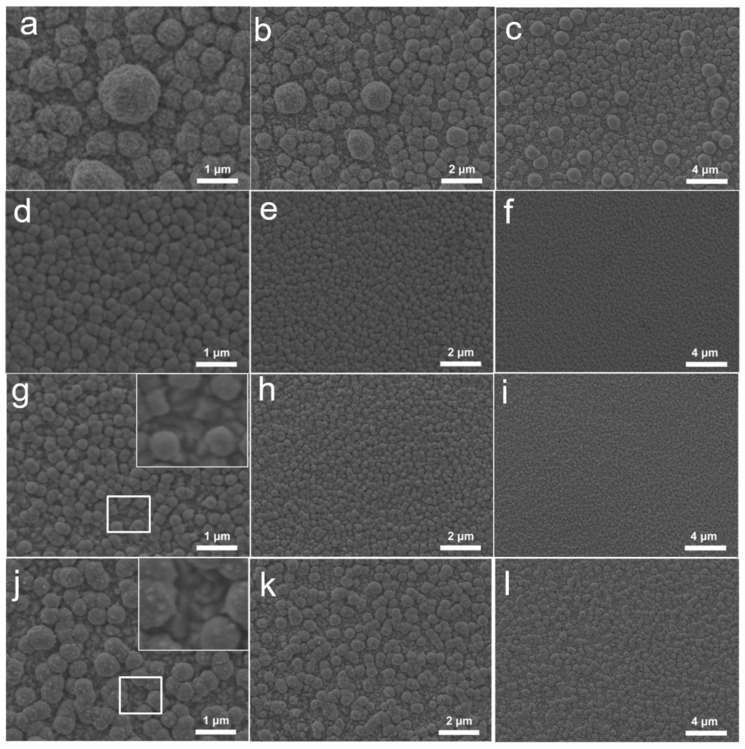
SEM images of the Cu-based electrodes with different scale bars for (**a**–**c**) Cu_0.3_, (**d**–**f**) Cu_0.6_, (**g**–**i**) Cu_0.9_, and (**j**–**l**) Cu_1.2_, inset images in (**g**,**j**) selected from the white squares, respectively.

**Figure 4 ijms-23-09373-f004:**
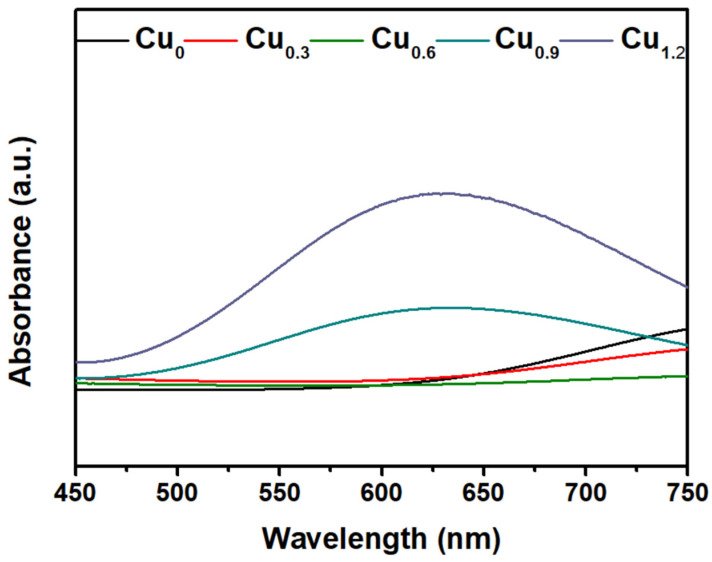
UV-vis absorption spectra of precursor solutions for Cu_0_, Cu_0.3_, Cu_0.6_, Cu_0.9_, and Cu_1.2_.

**Figure 5 ijms-23-09373-f005:**
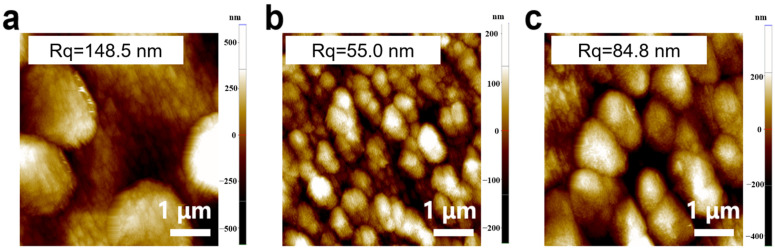
AFM images of copper-based electrodes: (**a**) Cu_0_, (**b**) Cu_0.6_, (**c**) Cu_1.2_.

**Figure 6 ijms-23-09373-f006:**
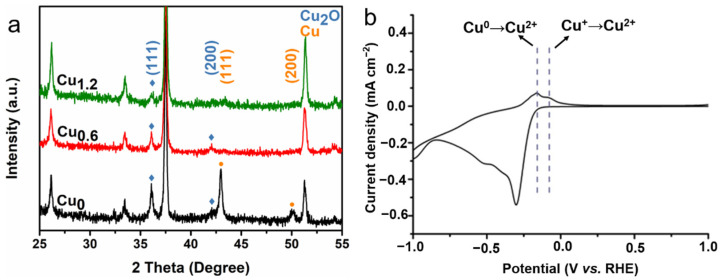
XRD patterns of the copper-based electrodes; unmarked peaks for FTO (**a**). CV for Cu_0.6_ as the anode (**b**).

**Figure 7 ijms-23-09373-f007:**
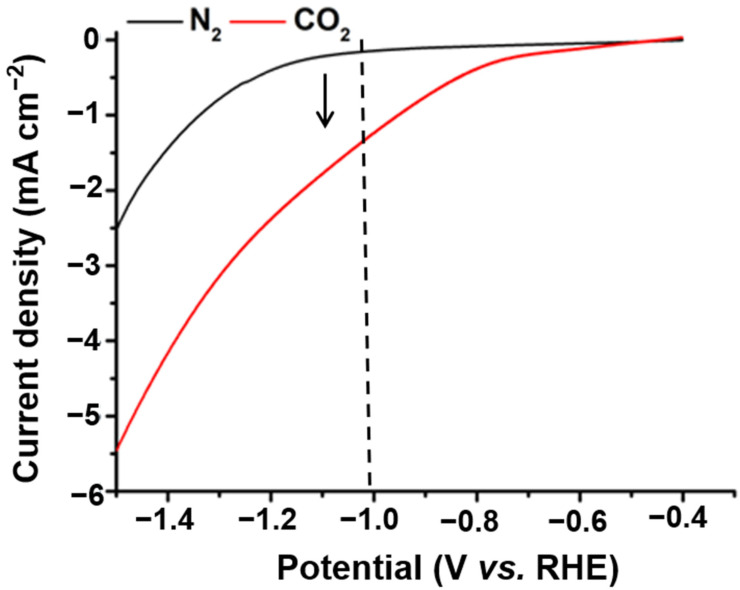
LSV curves of Cu_0.6_ in the 0.1 M KHCO_3_ solution saturated with N_2_ (black) and CO_2_ (red).

**Figure 8 ijms-23-09373-f008:**
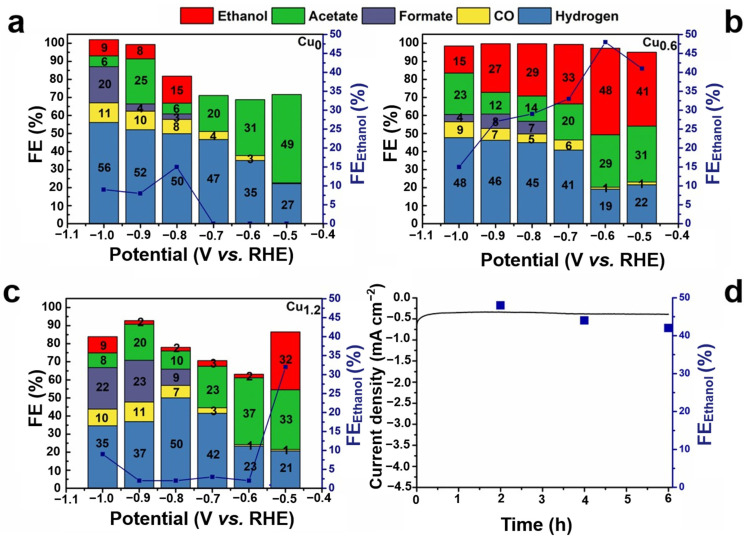
FEs of CO_2_RR products for the (**a**) Cu_0_, (**b**) Cu_0.6_, and (**c**) Cu_1.2_ electrodes under different potentials. (**d**) Current stability and FE of ethanol under −0.6 V vs. RHE potential for 6 h.

**Figure 9 ijms-23-09373-f009:**
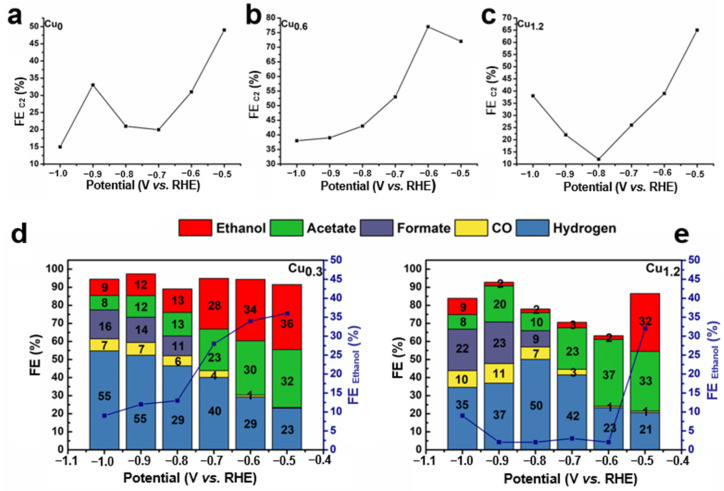
FE of C_2_ for (**a**) Cu_0_, (**b**) Cu_0.6_, and (**c**) Cu_1.2_ under different potentials. FE of each CO_2_ reduction products for (**d**) Cu_0.3_, (**e**) Cu_0.9_ under different potentials.

**Figure 10 ijms-23-09373-f010:**
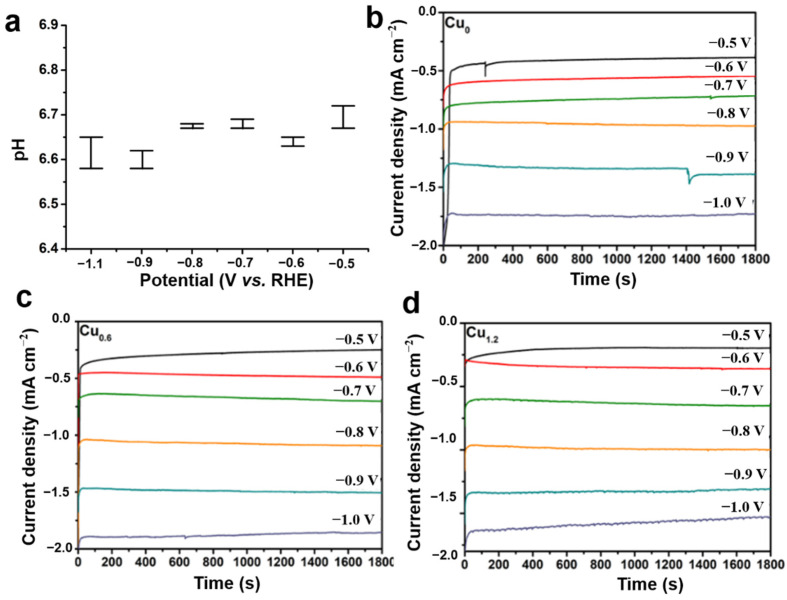
pH changes for Cu_0.6_ before and after the CO_2_RR (**a**), the standard deviation no more than 0.02. Current densities for (**b**) Cu_0_, (**c**) Cu_0.6_ and (**d**) Cu_1.2_ during CO_2_RR.

**Figure 11 ijms-23-09373-f011:**
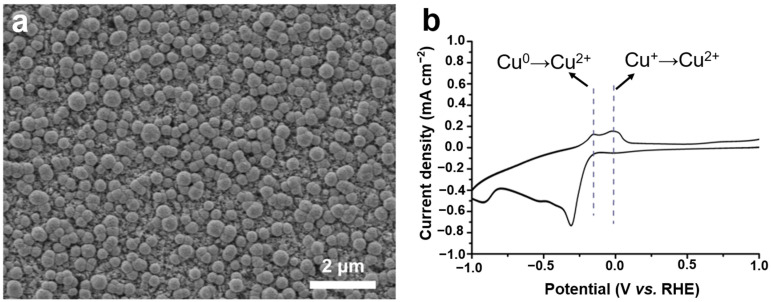
SEM images of Cu_0.6_ after the CO_2_RR (6 h) (**a**).CV for Cu_0.6_ as anode after CO_2_RR at −0.6 V vs. RHE (**b**).

**Figure 12 ijms-23-09373-f012:**
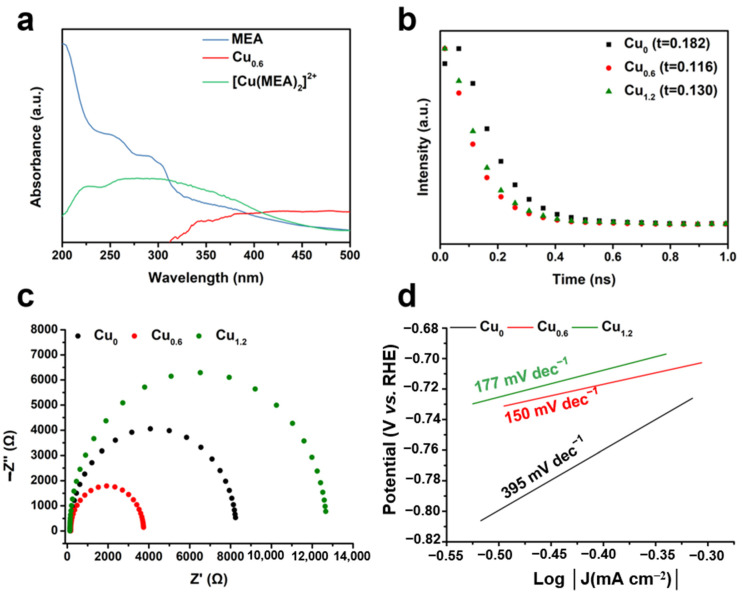
(**a**) UV-vis spectra of MEA (blue), Cu_0.6_ (red), and [Cu(MEA)_2_]^2+^ (green). (**b**) Fluorescence decay spectra of Cu_0_(black), Cu_0.6_(red), and Cu_1.2_(green). (**c**) Electrochemical impedance spectroscopy (EIS) of Cu_0_ (black), Cu_0.6_ (red), and Cu_1.2_ (green). (**d**) Tafel slopes of Cu_0_ (black), Cu_0.6_ (red), and Cu_1.2_ (green).

**Figure 13 ijms-23-09373-f013:**
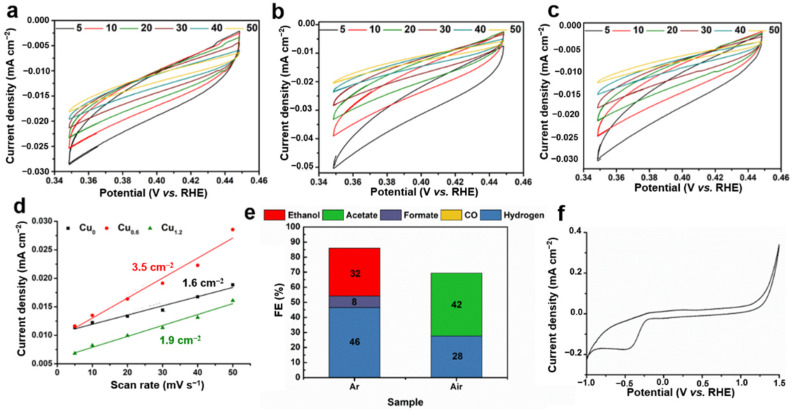
ECSA for (**a**) Cu_0_, (**b**) Cu_0.6_, (**c**) Cu_1.2_, and the electrochemical surface area for (**d**) Cu_0_ (black), Cu_0.6_ (red), and Cu_1.2_ (green). FE of the CO_2_RR products for Cu_0.6_ post-treated under different atmospheres at −0.6 V vs. RHE (**e**). CV of Cu_0.6_ post-treated in air at 150 °C for 6 h as the anode (**f**).

**Figure 14 ijms-23-09373-f014:**
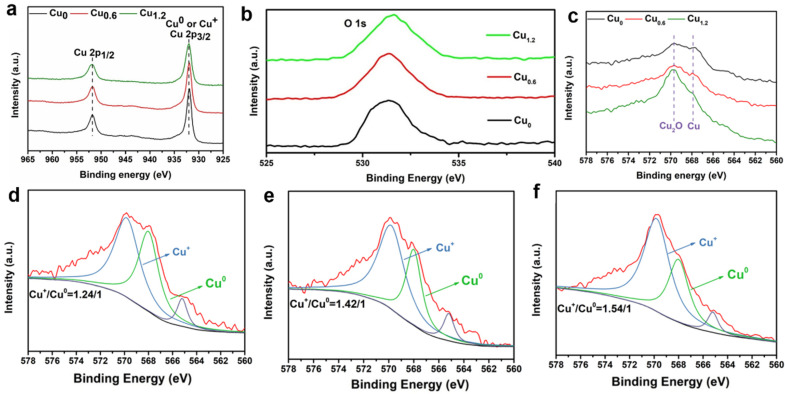
XPS peaks (**a**) Cu 2p and (**b**) O 1s of Cu_0_ (black), Cu_0.6_ (red), Cu_1.2_ (green). (**c**) Cu LMM Auger energy spectra of Cu_0_(black), Cu_0.6_(red), and Cu_1.2_(green) flacking powders. Cu 2p_3/2_ split peaks for (**d**) Cu_0_, (**e**) Cu_0.6_, and (**f**) Cu_1.2_.

**Figure 15 ijms-23-09373-f015:**
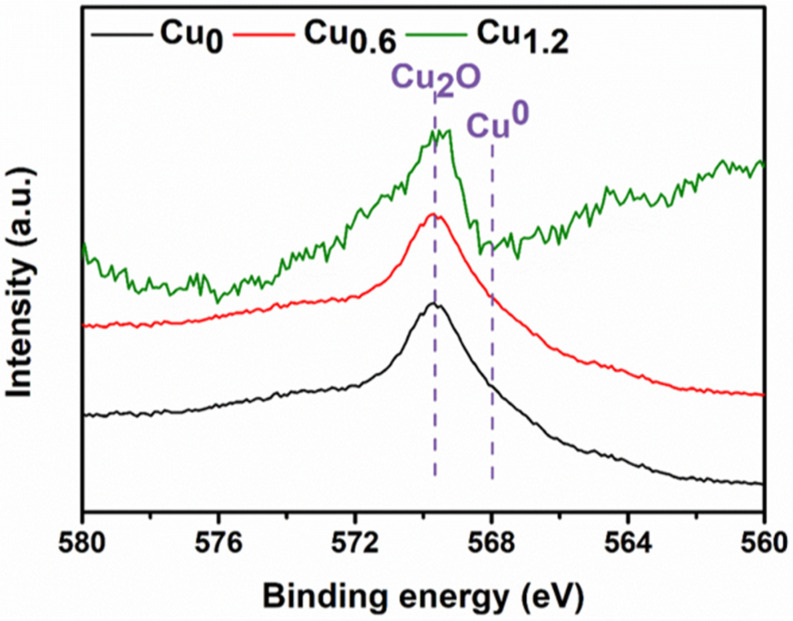
Cu LMM Auger energy spectra of Cu_0_ (black), Cu_0.6_ (red), Cu_1.2_ (green) electrodes.

**Figure 16 ijms-23-09373-f016:**
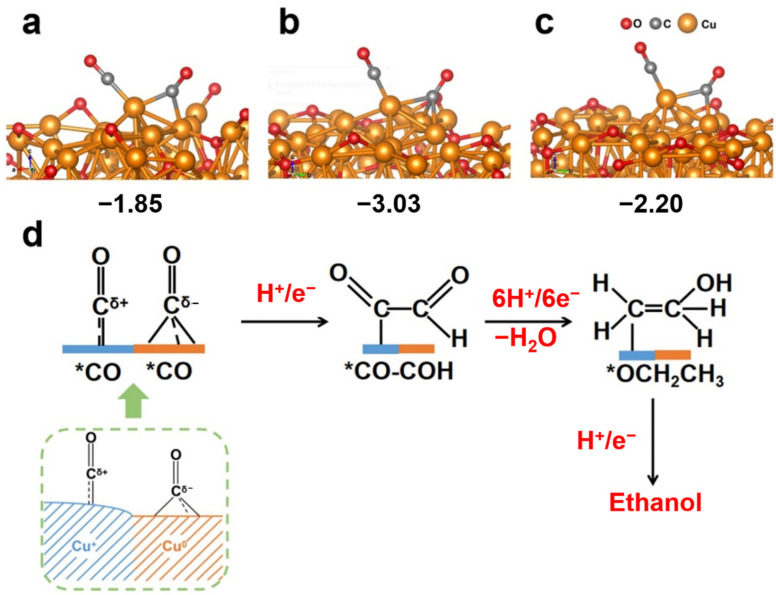
Free energy of *CO formation for (**a**) Cu_0_, (**b**) Cu_0.6_, and (**c**) Cu_1.2_. (**d**) Scheme of the reaction pathway of a CO_2_RR.

**Figure 17 ijms-23-09373-f017:**
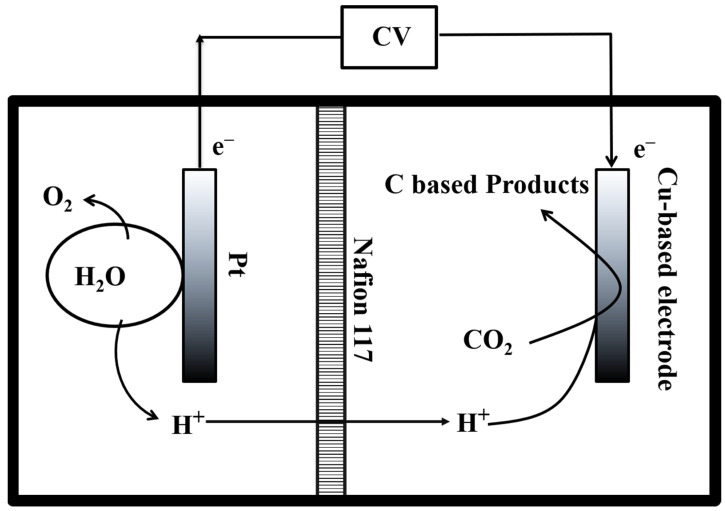
The scheme of CO_2_RR system for Cu-based electrodes.

**Figure 18 ijms-23-09373-f018:**
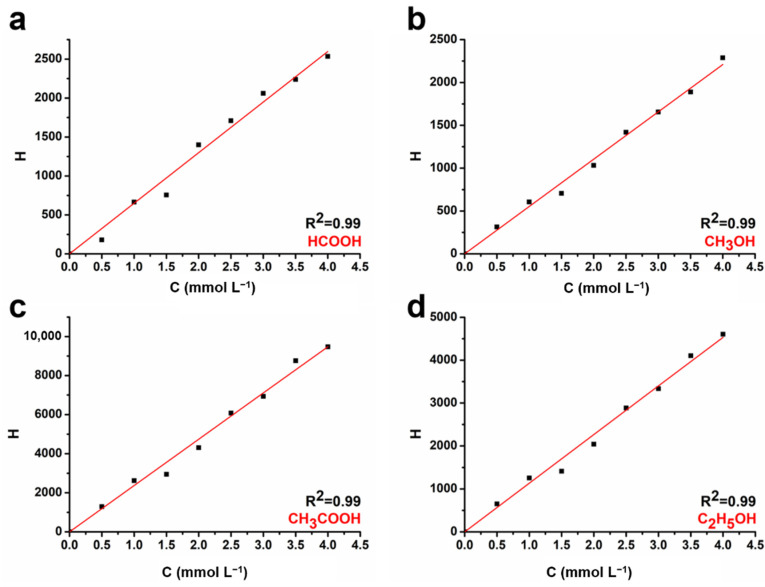
Standard curve lines of (**a**) HCOOH, (**b**) CH_3_OH, (**c**) CH_3_COOH, (**d**) C_2_H_5_OH.

**Table 1 ijms-23-09373-t001:** Comparison of ethanol and C_2_ FE from recent reports and in our work.

Catalyst	Electrolyte	FE (Ethanol)/%	FE (C_2_)/%	Ref.
Cu/Cu_2_O	0.1 M KCl	41.2%	81%	[12]
CuBr-DDT	0.5 M KCl	35.9%	72%	[13]
Cu/Cu_2_O-Ag-0.6	0.1 M KHCO_3_	19.2%	60.9%	[16]
SD-CuCd_2_	0.1 M KHCO_3_	32%	-	[17]
Cu_oh_-Ag	0.1 M KHCO_3_	23.1%	36.9%	[18]
Cu@Cu_2_O	0.1 M KHCO_3_	29%	50%	[25]
np-Cu@VO_2_-5%	0.1 M KHCO_3_	30.1%	-	[26]
Au_0.17_/Cu_2_O	0.1 M KHCO_3_	16.2%	-	[27]
Cu_0.6_	0.1 M KHCO_3_	48%	77%	our work

## Data Availability

The data supporting reported results are available on request from the corresponding author.

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
