# Peer review of "Concentration Optimization of Localized Cu0 and Cu+ on Cu-Based Electrodes for Improving Electrochemical Generation of Ethanol from Carbon Dioxide"

_ijms, 2022, doi:10.3390/ijms23169373_

Round 1
Reviewer 1 Report
Copper electrocatalysts were synthesized by electrodeposition frommonoethanolamine (MEA)-stabilized copper nitrate electrolytes, characterized, and tested for CO2 electroreduction. Significant emphasis was placed on the role of Cu0 andCu+ in the copper catalysts and the resultant ethanol faradaic efficiencies.The concentration ratio of Cu0 and Cu+ on the electrodes was precisely modulated by adding monoethanolamine (MEA). Among the different electrodes prepared based on Cu concentration, Cu0.6 electrode (containing 0.6% volume ratio of MEA in 0.1 mol/L of copper nitrate) exhibited flat surface and superior concentration ratios of Cu+ and Cu0 ions. This significantly improved the electrochemical efficiency and ethanol production of the CO2 reduction reaction (CO2RR).It provides good insights into fabrication of Cu-based electrodes and their performance in CO2RR.Irecommendthis study for acceptance by International Journal of Molecular Sciencesafter minor changes.
1. The “Figure 2” should be revised to “Figure 2a and b”in the line 125.
2. The exposed substrates of bared FTO are is difficult to be observed in Figures 3j-h and 3a-c. The enlarged SEM images is suggested to be provided.
3. “Table S1” should be revised to “Table 1” in the line 178.
4. “Figure 2d” should be revised to “Figure 8d” in the line 206.
5. The deviation of pH is suggested to be provided in Figure 10a.
6. For references of31, 32 and 33, the use of semicolonsis required to be unified.
Author Response
Point-To-Point Responses to Reviewers’ Comments
Reviewer #1 Review report "Concentration Optimization of Localized Cu0 and Cu+ on Cu-Based Electrodes for Improving Electrochemical Generation of Ethanol from Carbon Dioxide"
Comments: Copper electrocatalysts were synthesized by electrodeposition frommonoethanolamine (MEA)-stabilized copper nitrate electrolytes, characterized, and tested for CO2 electroreduction. Significant emphasis was placed on the role of Cu0 andCu+ in the copper catalysts and the resultant ethanol faradaic efficiencies.The concentration ratio of Cu0 and Cu+ on the electrodes was precisely modulated by adding monoethanolamine (MEA). Among the different electrodes prepared based on Cu concentration, Cu0.6 electrode (containing 0.6% volume ratio of MEA in 0.1 mol/L of copper nitrate) exhibited flat surface and superior concentration ratios of Cu+ and Cu0 ions. This significantly improved the electrochemical efficiency and ethanol production of the CO2 reduction reaction (CO2RR). It provides good insights into fabrication of Cu-based electrodes and their performance in CO2RR.Irecommendthis study for acceptance by International Journal of Molecular Sciencesafter minor changes.
Reponse: We thank the Reviewer for the following comments, which are very helpful for us to further improve our manuscript.
All changes are marked up using “Tracking Changes” in the manuscript.
- The “Figure 2” should be revised to “Figure 2a and b”in the line 125.
Reponse: We thank the Reviewer for his/her kind comments. “Figure 2” was revised to “Figure 2a and b”in the manuscript.
- The exposed substrates of bared FTO are is difficult to be observed in Figures 3j-h and 3a-c. The enlarged SEM images is suggested to be provided.
Reponse: We thank the Reviewer for his/her kind comments. The enlarged SEM images was provided in Figures 3j and 3a and the correlated illustration is added.
- “Table S1” should be revised to “Table 1” in the line 178.
Reponse: We thank the Reviewer for his/her kind comments. “Table S1” was revised to “Table 1”.
- “Figure 2d” should be revised to “Figure 8d” in the line 206.
Reponse: We thank the Reviewer for his/her kind comments. “Figure 2d” was revised to “Figure 8d” in the manuscript.
- The deviation of pH is suggested to be provided in Figure 10a.
Reponse: We thank the Reviewer for his/her kind comments. The deviation of pH was provided in the illustration of Figure 10a.
- For references of31, 32 and 33, the use of semicolonsis required to be unified.
Reponse: We thank the Reviewer for his/her kind comments. The references of 31, 32 and 33 were revised.
Reviewer 2 Report
1. chemical schemes of all electrochemical processes should be added
2. all LSV and CV should be pictorially described directly into all figure what the concrete peak or non peak signal means.
Author Response
Point-To-Point Responses to Reviewers’ Comments
Reviewer #2 Review report "Concentration Optimization of Localized Cu0 and Cu+ on Cu-Based Electrodes for Improving Electrochemical Generation of Ethanol from Carbon Dioxide"
All changes are marked up using “Tracking Changes” in the manuscript.
- chemical schemes of all electrochemical processes should be added
Reponse: We thank the Reviewer for his/her kind comments. The chemical schemes of electrochemical processes were added in the manuscript.
- all LSV and CV should be pictorially described directly into all figure what the concrete peak or non peak signal means.
Reponse: We thank the Reviewer for his/her kind comments. All the LSV and CV is pictorially described directly into all figure what the concrete peak or non peak signal means.